# Use of Yoghurt Enhanced with Volatile Plant Oils Encapsulated in Sodium Alginate to Increase the Human Body’s Immunity in the Present Fight Against Stress

**DOI:** 10.3390/ijerph17207588

**Published:** 2020-10-19

**Authors:** Ovidiu Tița, Maria Adelina Constantinescu, Mihaela Adriana Tița, Cecilia Georgescu

**Affiliations:** Department of Agricultural Sciences and Food Engineering, Lucian Blaga University of Sibiu, Doctor Ion Rațiu No.7, 550012 Sibiu, Romania; adelina.constantinescu@ulbsibiu.ro (M.A.C.); mihaela.tita@ulbsibiu.ro (M.A.T.); cecilia.georgescu@ulbsibiu.ro (C.G.)

**Keywords:** antioxidants, food management, mental health, stress, public health

## Abstract

(1) Background: The COVID–19 pandemic and the imposition of strict but necessary measures to prevent the spread of the new coronavirus have been, and still are, major stress factors for adults, children, and adolescents. Stress harms human health as it creates free radicals in the human body. According to various recent studies, volatile oils from various aromatic plants have a high content of antioxidants and antimicrobial compounds. An external supply of antioxidants is required to destroy these free radicals. The main purpose of this paper is to create a yoghurt with high antioxidant capacity, using only raw materials from Romania; (2) Methods: The bioactive components used to enrich the cow milk yoghurt were extracted as volatile oils out of four aromatic plants: basil, mint, lavender and fennel. Initially, the compounds were extracted to determine the antioxidant capacity, and subsequently, the antioxidant activity of the yoghurt was determined. The 2,2-diphenyl-1-picrylhy-drazyl (DPPH) method was used to determine the antioxidant activity; (3) Results: The results show that cow milk yoghurt enhanced with volatile oils of basil, lavender, mint and fennel, encapsulated in sodium alginate has an antioxidant and antimicrobial effect as a staple food with multiple effects in increasing the body’s immunity. The antioxidant activity proved to be considerably higher than the control sample. The highest antioxidant activity was obtained on the first day of the analysis, decreasing onwards to measurements taken on days 10 and 20. The cow milk yoghurt enriched with volatile basil oil obtained the best results; (4) Conclusions: The paper shows that yoghurts with a high antioxidant capacity were obtained, using only raw materials from Romania. A healthy diet, compliance with safety conditions and finding appropriate and safe methods to increase the body’s immunity is a good alternative to a major transition through harder times, such as pandemics. The creation of food products that include natural antioxidant compounds combines both the current great possibility of developing food production in Romania and the prevention and reduction of the effects caused by pandemic stress in the human body.

## 1. Introduction

According to the Food and Agriculture Organization (FAO) of the United States, agri-food production will increase by about 70% in the coming decades [1]. In 2019, Romania exported food and animals worth 4.77 billion euros, while imports were over 6.7 billion euros, according to data from the National Institute of Statistics. The crisis generated by the coronavirus could become an opportunity for Romania, especially in the area of food production, where it is a good time to invest in supporting the local chain of production, from small farms to processing plants. Romania currently imports a vast array of foods, from vegetables and fruits to meat, dairy products, sweets, bakery products or jams [2]. The satisfaction of requirements is not always present, the main purpose being the development and improvement of quality of life. These challenges require a change in industrial policy to increase the importance of the social, food and environmental components [3]. The extension of the term of validity of the food products, as well as the preservation of their appearance for a longer period, is an increasingly common condition among food processors. Unfortunately, mostly synthetic preservatives are used, in detriment to natural preservatives. Food preservatives are considered as a problem in the health of consumers. After numerous studies, it was concluded that the toxicity evolution of food preservatives should be tested, and that they should be subjected to pharmacological tests [4]. 

Numerous studies conducted in recent years have shown that regular consumption of dairy products can have a protective effect against the development of obesity and cardiovascular disease. The most consumed dairy products are yoghurt and cheese. Yoghurt is defined as a food in the form of a thick and slightly sour liquid that is obtained by adding bacteria in milk [5]. Yoghurt consumption has increased worldwide due to its nutritional value, therapeutic effects, and functional properties. To improve the nutritional and sensory properties of yoghurt, processors have begun to add different fruits or additives to their compositions [3]. Due to these characteristics, we decided to choose yoghurt as the dairy product that we will enrich with antioxidant compounds.

The COVID-19 pandemic and the imposition of strict but necessary measures to prevent the spread of the new coronavirus have been, and still are, a major stressor for adults, children and adolescents. During an epidemic, stress can have the following manifestations, according to experts from the Center for Disease Prevention and Control in the United States (CDC): fear and anxiety about one’s health and the health of loved ones, changes in sleeping or eating habits, alterations of sleep or concentration, worsening of chronic health problems and excessive use of alcohol, tobacco or other drugs [6]. 

To get over this period more easily and to reduce, as much as possible, the effects caused by the stress of the pandemic, doctors recommend adopting a healthier lifestyle. A healthy diet must be combined with daily sports training and spending as much time in nature as possible. In lack of a healthy and non-sedentary lifestyle, one can reach states of anxiety, depression or develop certain cardiovascular and mental illnesses [7]. 

The human immune system is very important for ensuring a normal life, which can prevent the mass spread of major health problems such as the emergence of novel viruses, like COVID-19 which has quickly become a global pandemic [8,9,10,11]. Remdesivir is used as a treatment, which acts on the ribonucleic acid (RNA) of the virus, preventing the virus from multiplying, being the only approved drug for the emergency treatment of coronavirus [12,13,14,15]. Another substance used in treatment is Lopinavir, a protease inhibitor, and its combination with Ritonavir has demonstrated excellent results in severe acute respiratory syndrome, in the initial phase, reducing the risk of death in patients [16]. After infection with the new coronavirus, the human body is weakened, prone to bacterial infections, with changes in physical parameters such as increased levels of C-reactive protein, neutrophilia, leukocytosis, alveolar lung opacity, increased serum procalcitonin [16,17,18]. Many plants in the Romanian flora have a pharmacological potential, which can be capitalized and used in medicine. This is an attempt to find viable solutions that would lead to the prevention of contamination of the body with various viruses by increasing the body’s immunity. In doing this, the body’s immunity can be a major first step in the fight against COVID-19 but also other vectors that can act negatively on the body. We chose basil, mint, lavender and fennel plants of continuous agricultural, medical and commercial interest for this study, due to the high content of active substances and other nutrients that are beneficial to human health. 

### 1.1. The Benefits of Aromatic Plants

After several types of research, it was discovered that the volatile oils extracted from different plants bring an extraordinary benefit to the health of consumers. These oils have antiseptic action especially on pathogenic bacteria such as *Listeria monocytogenes, Listeria innocua, Salmonella typhimurium*. Antioxidant activity is another benefit of volatile oils. Free radicals cause oxidation of biomolecules, including proteins, amino acids, deoxyribonucleic acid (DNA), etc., and eventually cause molecular changes related to ageing, arteriosclerosis and cancer [4].

Fennel (*Foeniculum vulgare*) is an aromatic plant that belongs to the *Apiaceae* family and is considered one of the oldest plants cultivated in the world. This is an annual, biennial or perennial plant native to the Mediterranean area grown for its aromatic fruits, which are used as culinary spices. It grows especially well in coastal climates and riverbanks [19,20]. Fennel essential oils are used as a flavoring agent in foods such as beverages, bread, pickles, pastries and cheese. It is also used as a component in cosmetics and pharmaceuticals. Fennel medicines and fennel essential oils have hepatoprotective effects as well as antispasmodic effects. Additionally, volatile fennel oil is known for its diuretic, anti-inflammatory, analgesic and antioxidant activity [21]. The antioxidant and antimicrobial activity of volatile fennel oil is offered by the high content of trans-anethole (63.30%), pinene (11.11%) and fenchone (8.32%) [20,21,22,23,24,25]

Basil (*Ocimum basilicum*) is an aromatic plant that belongs to the *Lamiaceae* family and is considered a rich source of essential oils. The basil plant is native to Asia, Africa, South and Central America [26]. Volatile basil oil has antimicrobial, antihistaminic, anti-inflammatory, anthelmintic, antioxidant properties, has an immunomodulatory effect, it is an antidepressant, antidiabetic and anti-hyperlipidemiac, has a hepatoprotective, neuroprotective and cardioprotective effect and it linked to anticancer activity [27]. Due to the content of estragole (41.40%), 1,6-octadien-3-ol, 3,7-dimethyl (29.49%), trans-alpha-bergamotene (5.32%), eucalyptol (3.51), citral (3.31%), N-Cyano-3-methylbut-2-enamine (3.08%), cis-alpha-bisabolene (1.92%), levomenthol (1.81%), and beta-myrcene (1.11%), volatile oil basil has a good antimicrobial and antioxidant activity [28,29,30,31,32,33].

Mint (*Mentha piperita*) is an aromatic plant that is part of the *Lamiaceae* family and is widely grown in Europe, Asia, Egypt, South Africa and Arabia [34]. Mint leaves are traditionally used as a tea in the treatment of headaches, fever, digestive disorders and various minor conditions. In modern medicine, mint is widely used in the treatment of gastrointestinal disorders [35]. Many studies that have evaluated the antioxidant activities of volatile mint oil have focused exclusively on chemical tests, while the effectiveness of volatile mint oil in preventing oxidative stress at the cellular level or in a living organism has not been characterized [36]. The main components of the mint volatile oil are menthol, menthone, menthofuran, isomenthone, (E)-caryophyllene, 1,8-cineole, linalool, limonene, carvone, pulegone and α-terpineol. They give the volatile mint oil an antioxidant and antimicrobial capacity [37,38,39].

Lavender (*Lavandula angustifolia*) is an aromatic plant that belongs to the *Lamiaceae* family. Lavender is a herbaceous plant native to the Mediterranean area and is widely cultivated. The smell of lavender improves mood, reduces mental stress and anxiety, and improves sleep [40]. Lavender essential oil is commonly used in aromatherapy and various complementary medicines and cosmetics [41]. Numerous studies have shown that volatile lavender oil has antioxidant, antimicrobial and antifungal properties [42]. The antioxidant and antimicrobial capacity of lavender volatile oil is offered by compounds such as the monoterpenoids, linalool, linalyl acetate, 1,8-cineole, β-ocimene, terpinen-4-ol, and camphor [43,44,45,46].

In recent years, Romanian entrepreneurs have focused on cultivating aromatic and medicinal plants. They focused especially on plants that are found in the Mediterranean area. If until a few years ago we met many crops of vegetables or plants specific to the Romanian area, in recent years many crops of lavender, fennel, mint or basil have appeared. The lavender culture is found in the plain and depression area of Romania because the plant needs a warm and moderately dry climate. The fennel culture is most often found in the southern part of Romania, where it is mostly plain and there is a warm climate. The mint culture is the most common in the depression area of Romania because it is a plant that has a moderate tolerance to drought, requiring numerous irrigations. Basil crops are most often found in southern Romania, more precisely in the plains because the plant prefers a warm climate. Basil is in first place at the top of Romanians’ preferences in terms of cultivating aromatic and medicinal plants [47,48,49,50,51].

In the study by Kokina et al. in 2019 lavender and peppermint volatile oils have been shown to have the highest antioxidant capacity. They combined two methods, DPPH and 2,2’-azino-bis-3-ethylbenzthiazoline-6-sulphonic acid (ABTS), to increase the efficiency of the evaluation of the antioxidant activity of the volatile oils studied. Volatile oils were stored for 12 months and a significant decrease in antioxidant activity was observed [52]. In the study conducted by Köksal and Gülçin in 2008 and 2010, they demonstrated that cauliflower and lintite extracts have strong antioxidant activity [53,54]. In 2016, Aslam et al. conducted a study that demonstrated the antioxidant capacity of spinach leaves [55]. Numerous studies conducted in recent years have shown that regular consumption of dairy products can have a protective effect against the development of obesity and cardiovascular disease [5]. Yoghurt consumption has increased worldwide due to its nutritional value, therapeutic effects and functional properties [20]. 

It is very important to know the morphology of the plant and its active substances. From this point of view, the selective use of useful compounds from the plant that can be directed exactly to obtain the expected effect is preferred, and less of the plants that have been proven to produce unwanted and sometimes toxic interactions [56]. The oils obtained, once analyzed and the useful components identified, offer the possibility of their exact dosage in yoghurt so that the antioxidant effect can be ensured without substantially changing the taste and smell of the product [57,58].

### 1.2. The Antioxidant Capacity of Aromatic Plants

Stress in the environment, especially heavy metal pollution, leads to the production of oxidative stress in plants. The population is constantly growing and the problems related to the supply of food are becoming more and more pressing. Finding viable solutions for the realization of basic food products, with the widest possible destination, which in addition to a longer life cycle can ensure at the same time a healthy lifestyle, is of utmost importance. Plants develop numerous enzymatic and non-enzymatic antioxidant mechanisms for detoxification. Aromatic plants are especially rich in antioxidant phenolic compounds. Their antioxidant activity is due to the redox properties and chemical structure, which play an important role in neutralizing free radicals and peroxides [20].

In a study conducted in 2019 on a sample of 1000 people in Romania, it was shown that 98% of them suffer from diseases caused by stress. Work related issues and the insecurity of tomorrow are the main reasons for the respondents’ anxiety [59]. As last year in Romania, the uncertainty of tomorrow and employment were the most widespread causes of stress, this year the world situation impacted by the COVID–19 crisis will only deepen this even more. The large numbers of illnesses reported in the country at the moment, as well as the severely affected economic situation, are the most important causes of stress. Social distancing, isolation and lack of certainty at work affects many people, who even reach states of anxiety and depression. Stress development is associated with an increase in the number of free radicals, a decrease in the levels of antioxidant enzymes and an increase in oxidative lipids in the brain tissues. This free radical activity is associated with impaired cognitive function. Major stress for a single period of eight hours increases the level of oxidative stress and the attack of free radicals on the brain, being accompanied by the weakening of memory and cognitive function. Antioxidant nutrients have been shown to alleviate these effects when administered before or after stress-induced circumstances [60].

Antioxidants are widely used as food additives to avoid food degradation. Antioxidants also play an important role in preventing a variety of lifestyle disorders and ageing conditions, as they are closely linked to active oxygen and lipid peroxidation [54]. Vegetable foods contain more antioxidants than those of animal origin, so the World Health Organization (WHO, Geneva, Switzerland) recommends about 400–600 g of vegetables and fruits daily to reduce the risk of cardiovascular disease, cancer, cognitive impairment and other eating disorders [61]. According to studies, it has been shown that the intake of antioxidants in the form of artificial supplements has not always brought the desired effect, so replacing them with natural antioxidants from plants can improve the desired effect. The intake of antioxidants, especially in this period when stress is present at maximum levels, is essential. The creation of food products that include natural antioxidant compounds combines both the current great possibility of developing food production in Romania and the prevention and reduction of the effects caused by pandemic stress in the human body.

To determine the antioxidant capacity, we used the DPPH method. The DPPH method measures the radical scavenging activity of antioxidants against free radicals, such as the DPPH radical [62].

## 2. Materials and Methods

### 2.1. Overview

The main purpose of this work is to create yoghurts with high antioxidant and antimicrobial capacity using only raw materials from Romania. External intake of antioxidants is essential to reduce the effects of daily stress. Volatile oils are an excellent source of antioxidants, and their use in food production can be a great direction for the current situation. Additionally, by using volatile oils, we aim to eliminate artificial preservatives added to yoghurts. According to studies in recent years, food preservatives harm consumer health, so using volatile oils with antimicrobial activity avoids the use of synthetic ones. To achieve this objective, we decided to enrich cow milk yoghurt with volatile oils encapsulated in sodium alginate. We used volatile oils from four aromatic plants: lavender, fennel, mint and basil. To achieve our purpose, we used the DPPH method to determinate antioxidant capacity, and the measurements were made on the first day after making yoghurt from cow’s milk, after 10 days and after 20 days. For each determination, we had two samples, a test sample (yoghurt with volatile oils) and a control sample (yoghurt without volatile oils). The yoghurt samples were packed in 150g plastic cups and stored in the refrigerator at a temperature between 0–4 °C. During the entire storage period, the glasses were covered with aluminum foil.

### 2.2. Materials

We used five samples, first called control sample (yoghurt in which no alginate capsules were added with volatile oil), the second one called the yoghurt sample with the addition of volatile mint oil, the third one called the yoghurt sample with volatile basil oil, the fourth one called the yoghurt sample with volatile fennel oil and the fifth one called the yoghurt sample with volatile lavender oil. We used the DPPH method to determinate antioxidant capacity, and the measurements were made on the first day after making yoghurt from cow milk, after 10 days and after 20 days. The cow’s milk was taken from a farm in the Sibiu area. Lavender, mint and basil were taken from a culture located in the Mureș area, and fennel was taken from a culture in the Ialomița area.

### 2.3. Procedure

In 2019—before deciding to make a food product with a high content of antioxidants—we conducted a market study and a sensory analysis for these types of yoghurts so that we can conclude whether the products made by us are to the liking of consumers. Following the sensory analysis, the tasters stated that the volatile oil added to the yoghurt only slightly influences its taste. The specific taste of yoghurt is predominant, only at the end, stimulating a slight taste of the plant from which the volatile oil is extracted. In addition to these two methods, we determined the pH and the lactic acid content to determine the antimicrobial activity of the yoghurt samples. To verify the antimicrobial activity of volatile oils, we created a mixture of the four oils studied and tested them on different cultures of enterobacteria, yeasts and molds. For the mixture of volatile oils, we used 25% volatile mint oil, 25% volatile lavender oil, 25% volatile basil oil and 25% volatile fennel oil. After obtaining the dilutions, the sowing took place on different culture mediums. The volatile oil mixture had a strong effect on colonies of enterobacteria, yeasts and molds: zero colony-forming units (CFU). No colony developed compared to the comparison plates (standard) [63].

To make the yoghurt enriched with bioactive components extracted from aromatic plants we used raw cow’s milk with a physico-chemical composition representative of the lactation period, which we pasteurized, then cooled it and added lactic crops. For the inoculation operation, we used a starter culture from Hansen (product name: F-DVS YC-X11 Yo-Flex). This is a thermophilic culture formed from *Lactobacillus delbrueckii subsp. bulgaricus* and *Streptococcus thermophillus*. The volatile oils did not influence the process of obtaining yoghurt. Volatile oil compounds are gradually released into yoghurt due to its encapsulation in sodium alginate. The gradual release of antimicrobial and antioxidant action of volatile oils ensures a longer time of action and an avoidance of losses.

For the extraction of volatile oils, we used mint, basil, lavender—dried and crushed aerial parts—and fennel seeds. The volatile oils were extracted by steam entrainment using the Neo-Clevenger apparatus modified by Moritz. The extraction time was five hours for each sample, and for efficient extraction, the plants were soaked the day before. At the end of the extraction, the volatile oil obtained was measured and 1 mL of benzene is added over it. It was then placed in a glass vial containing sodium sulfate anhydrous to remove any traces of water. Using a Pasteur pipette, we extracted the volatile oil from the glass vial and the benzene evaporated. To preserve the characteristics of the volatile oil until analysis, it was sealed in a dark ampoule and refrigerated.

The 4 samples of alginate capsules were obtained from 25 g 2% sodium alginate solution and 30 μL volatile basil, mint, fennel and lavender oil. The alginate solution was added gradually to the calcium chloride solution under centrifugation, thus obtaining the alginate capsules which were then washed with distilled water [63]. 

For the extraction of the compounds to determine the antioxidant capacity of the yoghurt samples with volatile oils, we used the extraction method adapted after Patel et al. (2016). We weighed 0.5 g of the sample to be analyzed and then extracted it with 10 mL of the mixture methanol:water:hydrochloric acid 0.12 M = 70:29:1 (v/v/v), at room temperature, for 24 h. The mixture was then kept on the ultrasonic water bath for 30 min at a temperature of 25 °C. After the time ran out, the supernatant was collected and centrifuged at 8000 rpm for 10 min. The residue was suspended in 10 mL of solvent to perform a second extraction for 15 min, on the bath of water at 25 °C. The resulting supernatant was centrifuged under the same conditions as the first. The total amount of supernatant was evaporated to the rotary evaporator and the residue was taken up with 10 mL of methanol. We filtered the mixture of supernatant and methanol and filled it to a volume of 10 mL with the same solvent [64]. 

To determine the antioxidant activity of the yoghurt samples with the addition of volatile oils encapsulated in sodium alginate, we used a method adapted according to the method applied by Tylkowski et al. (2011) for the ethanolic extracts of *Sideritis* ssp. L. We prepared a 25 µg/mL DPPH solution by solubilizing a quantity of DPPH in absolute methanol — stock solution.This mixture needed to be prepared in advance—at least 1–2 h—for complete solubilization. A volume of 970 µL DPPH solution 25 µg/mL is measured and added over 30 µL methanol extract from the samples to be analyzed, which we obtained using the extraction shown above. To interpret the results, we measured the absorbance at 515 nm for each sample using the CECIL 1021 UV-VIS spectrophotometer and the concentration is determined according to the standard curve obtained from different concentrations of the stock solution [65].

The determination of antioxidant activity was performed for each sample of yoghurt. For all these samples, there were 10 spectrophotometer readings, because from each type of yoghurt we made five samples (five containers of 100 g each). After extracting the necessary compounds to determine the antioxidant activity of each extracted container we took two readings. We wanted to eliminate all errors related to equipment, human error and differences in temperature or humidity. All readings were made on the same day of the analysis, to be more exact on the first day, on the 10th day and on the 20th day after the yoghurt samples were made.

## 3. Results

The results obtained in this research are presented as follows.

### 3.1. Control Sample—Simple Yoghurt in Which No Volatile Oil Capsules Were Added

Figure 1a shows the antioxidant activity of the control sample on the first day, the 10th day and the 20th day. On the first day of the analysis, the highest values of the antioxidant activity of the control sample were obtained. On this day, the maximum value obtained was 0.23%, and the minimum value was 0.16%. The value of the antioxidant activity for the control sample decreased on the 10th day compared to the first day. On day 10 of the analysis, the highest value obtained was 0.16%, and the lowest value was 0.10%. The lowest values of antioxidant activity for the control sample were obtained on day 20 of the analysis. On this day, the highest measured value was 0.11%, and the lowest was 0.06%. The average value for each day was 0.20% for the first day, 0.13% for the 10th day and 0.08% for the 20th day. 

In Figure 1a, the decline is calculated on day 10 and day 20 compared to the first day for the control sample. The decline on day 10 compared to the first day is smaller than the decline on day 20 compared to the first day. The results obtained on day 20 are lower compared to day 10. The decline from day 10 compared to the first day is between 52–88%. The decline from day 20 compared to the first day is between 27–58%. 

### 3.2. Cow Milk Yoghurt with the Addition of Volatile Mint Oil Encapsulated in Sodium Alginate

Figure 2a shows the antioxidant activity of the yoghurt sample from cow milk with the addition of volatile mint oil encapsulated in sodium alginates on the first day, the 10th day and the 20th day. The strongest antioxidant activity of the yoghurt sample with volatile mint oil was on the first day. The lowest value of this day was 1.57%, and the highest was 1.62%. The value of the antioxidant activity decreased on the 10th day compared to the first day. On day 10 the highest value was 1.23%, and the lowest at 1.17%. For this yoghurt sample, the lowest antioxidant activity was recorded on day 20. The highest value on day 20 was 1.06%, and the lowest was 0.95%. The average value for each day is 1.60% for the first day, 1.20% for the 10th day and 1.00% for the 20th day. 

In Figure 2b, the decline is calculated on day 10 and day 20 compared to the first day for the yoghurt sample from cow milk with the addition of volatile mint oil encapsulated in sodium alginates. The decline on day 10 compared to the first day is smaller than the decline on day 20 compared to the first day. The results obtained on day 20 are lower compared to day 10. The decline from day 10 compared to the first day is between 73–78%. The decline from day 20 compared to the first day is between 61–66%.

### 3.3. Cow Milk Yoghurt with the Addition of Volatile Basil Oil Encapsulated in Sodium Alginate

Figure 3a shows the antioxidant activity of the yoghurt sample from cow’s milk with the addition of volatile basil oil encapsulated in sodium alginates on the first day, the 10th day and the 20th day. On the first day, the yoghurt sample with volatile basil oil showed the highest antioxidant activity, and on the 20th the lowest. The highest value from the first day was 9.65% and the lowest at 9.56%. On day 10 the highest value was 9.33%, and the lowest at 9.26%. On day 10 the antioxidant activity was lower than on the first day, but it was higher than on day 20. Day 20 showed the lowest antioxidant activity, and the lowest value was 8.65%. The average value for each day is 9.60% for the first day, 9.30% for the 10th day and 8.70% for the 20th day.

In Figure 3b, the decline is calculated on day 10 and day 20 compared to the first day for the yoghurt sample from cow milk with the addition of volatile basil oil encapsulated in sodium alginates. The decline on day 10 compared to the first day is smaller than the decline on day 20 compared to the first day. The results obtained on day 20 are lower compared to day 10. The decline from day 10 compared to the first day is between 96–97%. The decline from day 20 compared to the first day is between 90–91%.

### 3.4. Cow Milk Yoghurt with the Addition of Volatile Fennel Oil Encapsulated in Sodium Alginate

Figure 4a shows the antioxidant activity of the yoghurt sample from cow milk with the addition of volatile fennel oil encapsulated in sodium alginates on the first day, the 10th day and the 20th day. The highest antioxidant activity was on the first day and the lowest on the 20th day. On the first day, the highest value was 6.43%, and the lowest value was 6.37%. The antioxidant activity on day 10 is lower than on the first day. The highest value on the 10th day was 6.18%, and the lowest was 6.22%. On day 20, the yoghurt sample with volatile fennel oil had the lowest antioxidant activity. The highest value recorded was 6.04%, and the lowest was 5.95%. The average value for each day was 6.40% for the first day, 6.20% for the 10th day and 6.00% for the 20th day.

In Figure 4b, the decline is calculated on day 10 and day 20 compared to the first day for the yoghurt sample from cow milk with the addition of volatile fennel oil encapsulated in sodium alginates. The decline on day 10 compared to the first day is smaller than the decline on day 20 compared to the first day. The results obtained on day 20 are lower compared to day 10. The decline from day 10 compared to the first day was between 96–97%. The decline from day 20 compared to the first day was between 93–95%.

### 3.5. Cow Milk Yoghurt with the Addition of Volatile Lavender Oil Encapsulated in Sodium Alginate

Figure 5a shows the antioxidant activity of the yoghurt sample from cow milk with the addition of volatile lavender oil encapsulated in sodium alginates on the first day, the 10th day and the 20th day. The highest antioxidant activity of the yoghurt sample with volatile lavender oil was on the first day, and the lowest was on the 20th day. The highest value on the first day was 5.23%, and the lowest was 5.17%. The antioxidant activity on the 10th day was lower than on the first day, but it was higher than on the 20th day. The highest value on day 10 was 5.13%, and the lowest was 5.07%. On day 20, the lowest antioxidant activity was recorded, and the lowest value was 4.67%. The average value for each day was 5.20% for the first day, 5.10% for the 10th day and 4.70% for the 20th day.

In Figure 5b, the decline is calculated on day 10 and day 20 compared to the first day for the yoghurt sample from cow milk with the addition of volatile lavender oil encapsulated in sodium alginates. The decline on day 10 compared to the first day is smaller than the decline on day 20 compared to the first day. The results obtained on day 20 are lower compared to day 10. The decline from day 10 compared to the first day was between 97–99%. The decline from day 20 compared to the first day was between 89–91%.

## 4. Discussion

The determination of the antioxidant activity was performed for each sample of yoghurt. For all samples, 10 spectrophotometer readings were performed to eliminate all errors related to equipment, human error and differences in temperature or humidity. All readings were made on the same day of the analysis, to be more exact on the first day, on the 10th day and on the 20th day after the yoghurt samples were made. 

According to Figure 6, the analytical results obtained show that the highest antioxidant activity was shown by the sample of yoghurt with basil volatile oil encapsulated in sodium alginate, followed by the sample of yoghurt with fennel volatile oil encapsulated in sodium alginate, then the sample of yoghurt with lavender volatile oil encapsulated in the sodium alginate and yoghurt sample with mint volatile oil encapsulated in sodium alginate. The antioxidant activity proved to be considerably higher than the control sample. 

The highest antioxidant activity was obtained on the first day of the analysis, before decreasing on day 10 and day 20. In the case of yoghurt samples with volatile basil, fennel and lavender oil, the greatest decreases in antioxidant activity were recorded between the 10th and the 20th day. In the case of the yoghurt sample with volatile mint oil, the greatest decrease in antioxidant activity was recorded between the first and the 10th day. These measurements show us that volatile basil, fennel and lavender oils have a higher antioxidant activity and are more stable during the first 10 days of preservation. In the case of volatile mint oil, it has a lower antioxidant activity and begins to stabilize after 10 days of preservation.

All the data obtained show us that the chosen product has a significant antioxidant capacity and can be used as an external source of antioxidants. It was thus demonstrated that there exists a way of increasing both the food and nutritional quality of the yoghurt but also the validation of the method of increasing the shelf life of the yoghurt.

## 5. Conclusions

Acid dairy products are appreciated worldwide because of the benefits they bring to the consumer’s health, as well as the possibility of consuming them from an early age [5]. In this case, we used yoghurt as a representative product for consumers, both in terms of favorable intake for the harmonious development of the body at different stages of life, but also in terms of frequency of consumption. Enriching it with bioactive components extracted from native herbs mint, basil, fennel and lavender, respectively, by adding volatile oils extracted from these plants and encapsulated using 2% sodium alginate, proved to be a beneficial option to increase the value of the product. The use of sodium alginate capsules also solved the problem of ensuring the stability of the volatile oils added to the food. The statistically processed results demonstrate the validity of the method of obtaining valuable food products, enriched in bioactive components, respectively, of using volatile oils with antioxidant and antimicrobial activity. The amount of volatile oils extracted from mint, basil, fennel and lavender is dependent on the growing conditions, soil and climatic conditions, the extraction method used, but the average values obtained support the potential for their use, especially in terms of the benefits for consumers’ health. 

The creation of foods containing natural antioxidant and antimicrobial compounds must take precedence in food management. The problem of daily stress has become a major health problem in recent years, and the global situation impacted by the COVID–19 crisis deepens this further. It is a reality nowadays that social distancing, isolation and the lack of certainty at work affects many people, even reaching states of anxiety and depression. Making such foods to increase the body’s immunity to viruses or to alleviate many chronic health problems is an effective and safe alternative to ensuring physical and mental health.

To get over this period more easily and to reduce—as much as possible the effects—caused by the stress resulted from the pandemic, doctors recommend adopting a healthier lifestyle. Therefore, a healthy diet, compliance with safety conditions and especially finding appropriate and safe methods to increase the body’s immunity are safe alternatives to an easier passage through harder periods such as the pandemic.

The use of volatile oils also ensures the complete elimination of artificial preservatives added to dairy products in this case of yoghurts. Food preservatives harm consumer health, often causing food poisoning, so using volatile oils with antiseptic and antioxidant activities ensures an increase in the shelf life of food and attention to toxicology and public health. 

## Figures and Tables

**Figure 1 ijerph-17-07588-f001:**
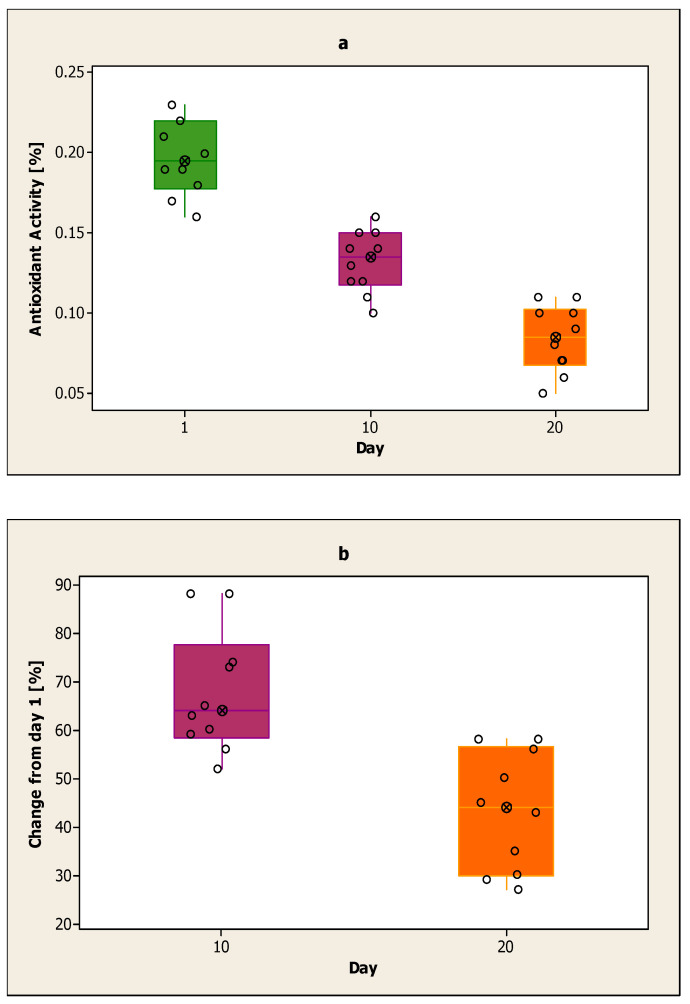
(**a**) Antioxidant activity of the control sample on the first day, the 10th day and the 20th day; (**b**) Decline on day 10 and day 20 for the control sample compared to day one.

**Figure 2 ijerph-17-07588-f002:**
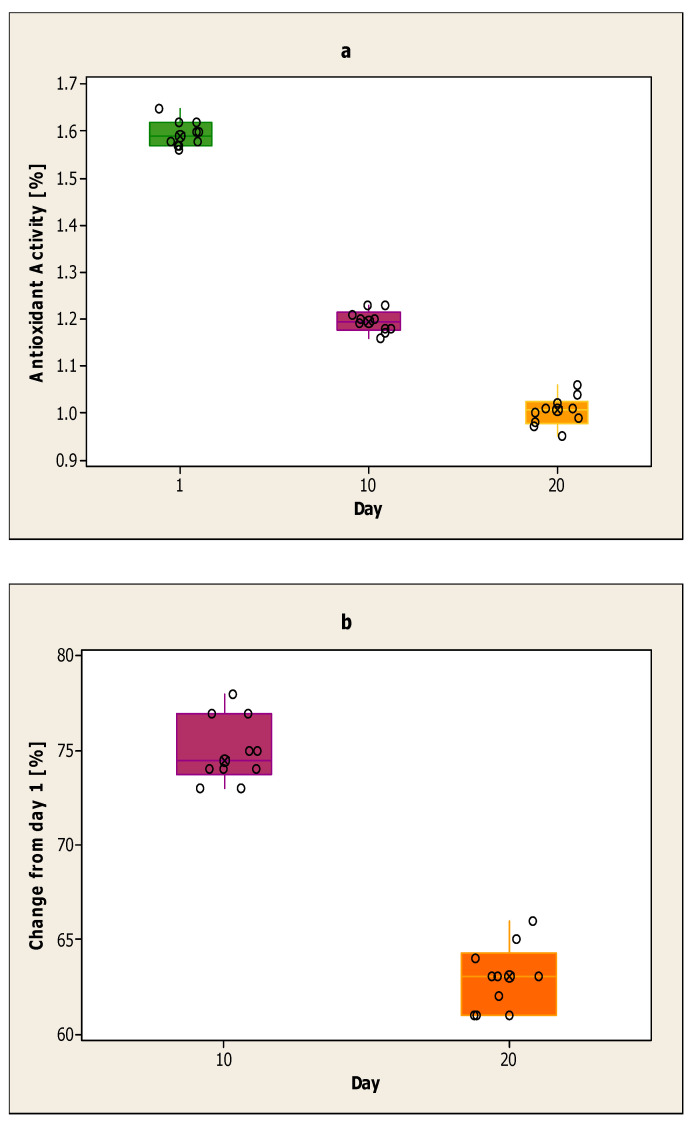
(**a**) Antioxidant activity of the yoghurt sample from cow milk with the addition of volatile mint oil encapsulated in sodium alginates on the first day, the 10th day and the 20th day; (**b**) Decline on day 10 and day 20 for the yoghurt sample from cow’s milk with the addition of volatile mint oil encapsulated in sodium alginates compared to day one.

**Figure 3 ijerph-17-07588-f003:**
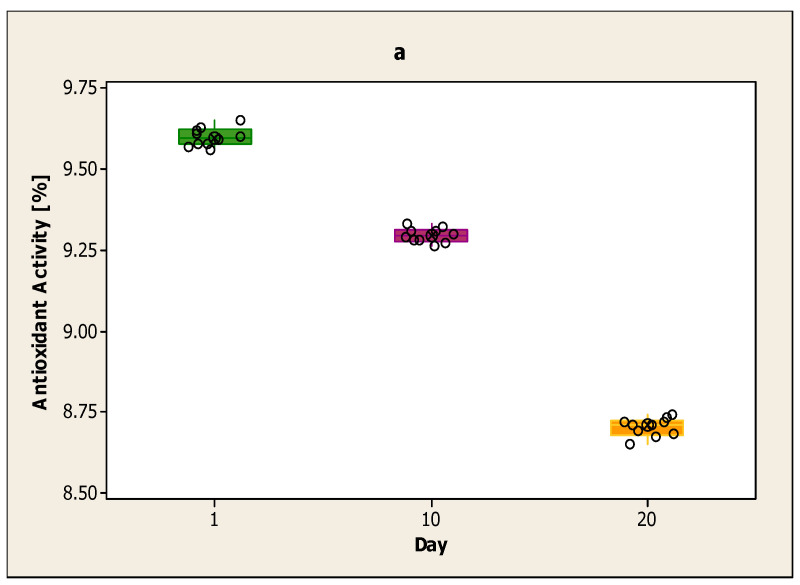
(**a**) Antioxidant activity of the yoghurt sample from cow milk with the addition of volatile basil oil encapsulated in sodium alginates on the first day, the 10th day and the 20th day; (**b**) Decline on day 10 and day 20 for the yoghurt sample from cow milk with the addition of volatile basil oil encapsulated in sodium alginates compared to day one.

**Figure 4 ijerph-17-07588-f004:**
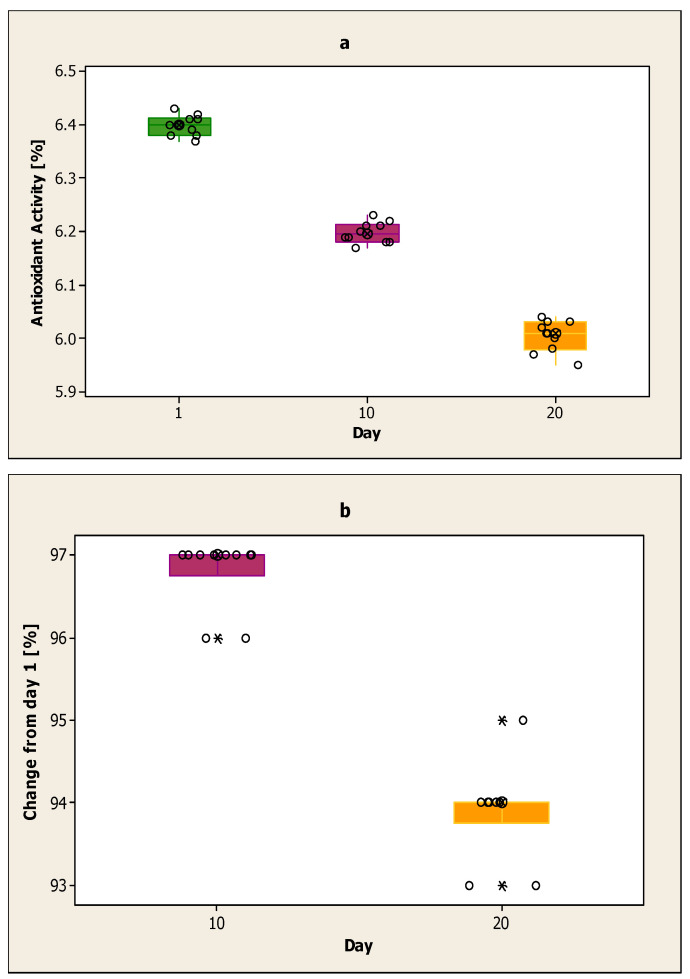
(**a**) Antioxidant activity of the yoghurt sample from cow milk with the addition of volatile fennel oil encapsulated in sodium alginates on the first day, the 10th day and the 20th day; (**b**) Decline on day 10 and day 20 for the yoghurt sample from cow milk with the addition of volatile fennel oil encapsulated in sodium alginates compared to day one.

**Figure 5 ijerph-17-07588-f005:**
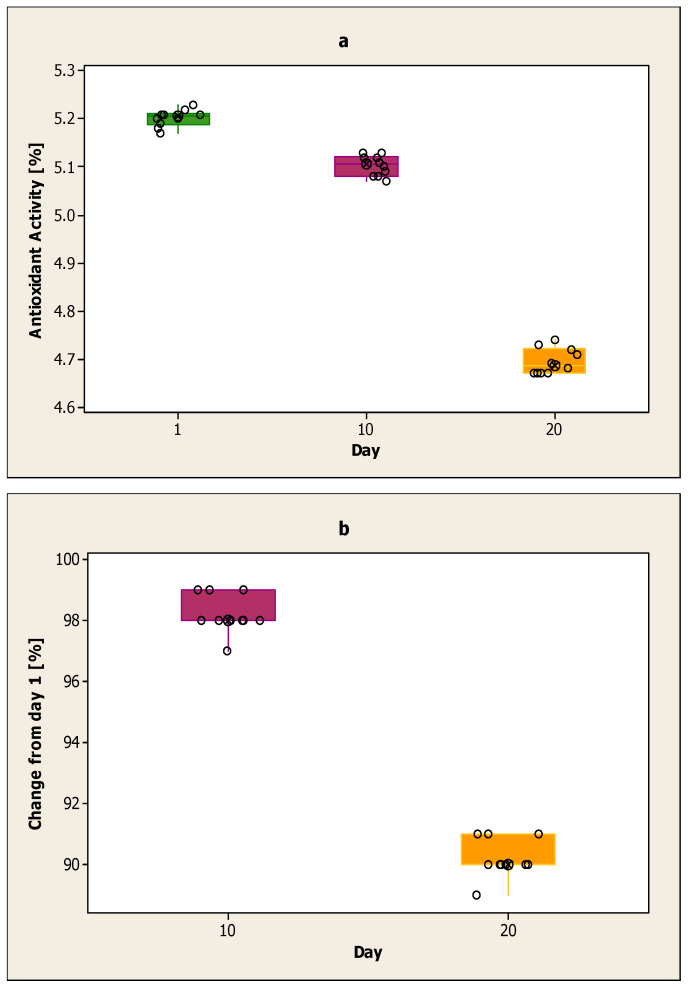
(**a**) Antioxidant activity of the yoghurt sample from cow milk with the addition of volatile lavender oil encapsulated in sodium alginates on the first day, the 10th day and the 20th day; (**b**) Decline on day 10 and day 20 for the yoghurt sample from cow milk with the addition of volatile lavender oil encapsulated in sodium alginates compared to day one.

**Figure 6 ijerph-17-07588-f006:**
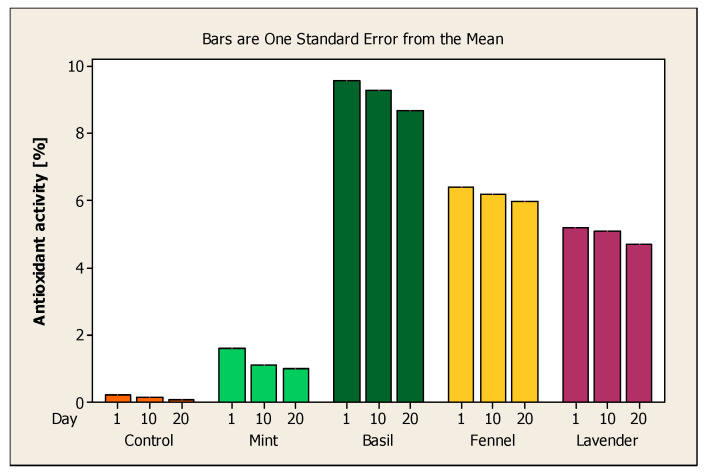
The average antioxidant activity of the yoghurt samples on the first day, the 10th day and the 20th day of analysis.

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
