# Peer review of "Use of Yoghurt Enhanced with Volatile Plant Oils Encapsulated in Sodium Alginate to Increase the Human Body’s Immunity in the Present Fight Against Stress"

_ijerph, 2020, doi:10.3390/ijerph17207588_

Round 1
Reviewer 1 Report
As stated in the article, the primary aim was to create food products with high antioxidant and antimicrobial capacity using only raw materials from the area of Romania. The authors added a series of volatile oils to cow’s milk yogurt and the main finding was that basil volatile oil encapsulated in sodium alginate produced the highest antioxidant activity.
I have several concerns with this article:
-The title of the article in no way reflects the research and findings presented in it. The title should be changed.
-The article needs extensive language revisions.
-Abstract: Please be specific with the main findings.
Introduction: Although the authors stated the aim as the creation of “food products”, which implies multiple food products, they only enriched yogurt. Please be specific in stating the primary aim of the project.
Results: Figures are low quality/not legible. In addition, the presentation of results is very cumbersome. I would recommend making multiple panel figures to present results in a more efficient manner. As presented, there are just too many figures/tables that can be combined. Figure 22 is lacking a y-axis label.
Author Response
Response to Reviewer 1 Comments
Point 1: The title of the article in no way reflects the research and findings presented in it. The title should be changed.
Response 1: We change the title of the article to “Use of Yoghurt Enhanced with Volatile Plant Oils Encapsulated in Sodium Alginate to Increase the Human Body's Immunity in the Present Fight Against Stress”. According to your specifications, we decided to be more precise from the title of the article and to mention the studied food product, the yoghurt enhanced with volatile plant oils. Please see: Row 1–5.
Point 2: The article needs extensive language revisions.
Response 2: Following your recommendations, we rewrote a large part of the article and corrected the initial mistakes. We followed the recommendations in the English language guidelines for submissions to MDPI journals.
Point 3: Abstract: Please be specific with the main findings
Response 3: In the abstract, we rewrote the results and some of the conclusions. We added more details related to the results obtained in the case of the volatile oil samples and the control sample. We also specified in which analysis period the best results were obtained and which yoghurt sample had the highest antioxidant activity. Please see: Row 23–35.
Point 4: Introduction: Although the authors stated the aim as the creation of “food products”, which implies multiple food products, they only enriched yogurt. Please be specific in stating the primary aim of the project.
Response 4: In the abstract, introduction and materials and methods we specified that it is only about yoghurt and not about several types of food products. Only in the introduction, we left a few more details related to global and local food production. In the introduction, we added some details about yoghurt and the health benefits to the consumer. Please see: Row 54–61.
Point 5: Results: Figures are low quality/not legible. In addition, the presentation of results is very cumbersome. I would recommend making multiple panel figures to present results in a more efficient manner. As presented, there are just too many figures/tables that can be combined. Figure 22 is lacking a y-axis label.
Response 5: We gave up almost all the figures. We decided to leave for each yoghurt sample the boxplot graph marked with (A) in which we highlighted the evolution of antioxidant activity in the three days of analysis. In the boxplot figure marked with (B), we have an obvious decline in antioxidant activity on days 10 and 20 compared to the first day of analysis. We put the two graphs in parallel for each yoghurt sample and analysed their antioxidant activity. Please see: Row 295–401.
Figure 22 is now the new Figure 6. We changed the type of graphical representation and added the units of measurement on the Y-axis label. Please see: Row 408.
Thank you very much for your constructive comments and insightful suggestions on the manuscript.

Reviewer 2 Report
Tita O. et al investigate the anti-oxidant properties of yoghurt fortified with oils extracted from Mint, Basil, Lavender and Fennel. For each series of measurements, five cultures of each herb were created, as well as a control culture. Each of the five cultures for each of the five conditions was measured twice on day 1, day 10 and day 20 using the DPPH method.
Substantial revision is required to bring this manuscript to a scholarly standard. Much of what is presented is unnecessary. For example data is presented as tables, box plots and histograms; one form is sufficient. Most of statistical explanations are unnecessary. The connection with Covid-19 is tenuous at best. Many details are not necessary e.g. breeds of cows in Romania. Roughly 2/3rds of this submission could be cut without loss.
Strongly suggested revisions.
1 Figures.
Cut all figures except Figures 1, 5, 9, 13, 17.
These figures are now Figure 1A, 2A, 3A, 4A, 5A.
For Figures 1B, 2B, 3B, 4B, 5B calculate the % decline on day 10 and day 20 for each culture compared to day 1. Plot this as a bar graph or boxplot next to each respective 'A' figure; there will be 2 bars/plots per graph.
New Figure 6 is the current Figure 22.
On new Figure 6 make the following changes -
Add error bars (either SEM or SD),
Add a Y-axis label including units
Modifications have been added as an attached file.
2 Analysis
Remove all reference to normality of the distribution, aberrant (?) data, kurtosis, skewedness etc. It is unnecessary.
The results of a 2 way ANOVA with Tukey post-hoc analysis (GraphPad Prism) is included in the attached file. At each timepoint, there is a significant difference between each condition.
3 Manuscript.
The manuscript needs a substantial re-write to remove extraneous text.
Can the authors comment on the activity of essential oils from these ubiquitous herbs on the microbes in the yoghurt? A reduction in CFU?
Can the authors suggest why yoghurt+oils is better than simply eating the herbs?
Does the addition of herb oils reduce the palatability of the yoghurt?
The units of 'anti-oxidant activity' are not explained sufficiently.
The method of oil extraction and anti-oxidant analysis are not explained sufficiently.
How were the cultures stored over 20 days?

Author Response
Response to Reviewer 2 Comments
Point 1: 1 Figures. Cut all figures except Figures 1, 5, 9, 13, 17. These figures are now Figure 1A, 2A, 3A, 4A, 5A.
For Figures 1B, 2B, 3B, 4B, 5B calculate the % decline on day 10 and day 20 for each culture compared to day 1. Plot this as a bar graph or boxplot next to each respective 'A' figure; there will be 2 bars/plots per graph.
New Figure 6 is the current Figure 22. On new Figure 6 make the following changes -Add error bars (either SEM or SD). Add a Y-axis label including units
Response 1: Following your recommendations, we cut all figures except old Figures 1, 5, 9, 13, 17. Now, these figures are Figure 1 (A), 2 (A), 3 (A), 4 (A), 5 (A). In these figures, we have highlighted the evolution of antioxidant activity in the three days of analysis.
For Figures 1 (B), 2 (B), 3 (B), 4 (B), 5 (B) we calculated the decline on day 10 and day 20 for each sample compared to day one. We put the two graphs in parallel for each yoghurt sample and analysed their antioxidant activity. Please see: Row 295–401.
Figure 22 is now the new Figure 6. We added on this error bars and Y-axis label including units. Please see: Row 408.
We made all the graphics in the Minitab program.
Point 2: 2 Analysis. Remove all reference to normality of the distribution, aberrant (?) data, kurtosis, skewedness etc. It is unnecessary. The results of a 2 way ANOVA with Tukey post-hoc analysis (GraphPad Prism) is included in the attached file. At each timepoint, there is a significant difference between each condition.
Response 2: Following your recommendations, we removed all reference to the normality of the distribution, aberrant data, kurtosis, skewedness etc. In interpreting the results we highlighted the evolution of antioxidant activity on each day of analysis, the decline on day 10 and day 20 compared to the first day and we indicated the maximum and minimum values recorded for each yoghurt sample. Please see: Row 293-418.
Point 3: 3 Manuscript.
a.The manuscript needs a substantial re-write to remove extraneous text.
b.Can the authors comment on the activity of essential oils from these ubiquitous herbs on the microbes in the yoghurt? A reduction in CFU?
c.Can the authors suggest why yoghurt+oils is better than simply eating the herbs?
d.Does the addition of herb oils reduce the palatability of the yoghurt?
e.The units of 'anti-oxidant activity' are not explained sufficiently.
f.The method of oil extraction and anti-oxidant analysis are not explained sufficiently.
g.How were the cultures stored over 20 days?
Response 3:
a.Following your recommendations, we rewrote a large part of the article and corrected the initial mistakes. We followed the recommendations in the English language guidelines for submissions to MDPI journals.
b.The volatile oils did not influence the process of obtaining yoghurt. The technological process of obtaining the yoghurt took place identically for the control sample and the samples with volatile oils. Volatile oil compounds are gradually released into yoghurt due to its encapsulation in sodium alginate. The properties of volatile oils begin to be gradually released into yoghurt after the first day of storage. These things were presented in a paper developed and published last year. In this paper, we highlighted the antimicrobial capacity of volatile oils and their action on yoghurt samples (determination of lactic acid content and pH). Also in this paper, we performed the sensory analysis of yoghurt samples. The work we have specified now can be found in the bibliography as number 57. Please see: 247–250.
To verify the antimicrobial activity of volatile oils (a reduction in CFU), we created a mixture of the four oils studied and tested them on different cultures of enterobacteria, yeasts and moulds. For the mixture of volatile oils, we used 25% volatile mint oil, 25% volatile lavender oil, 25% volatile basil oil and 25% volatile fennel oil. After obtaining the dilutions, the sowing took place on different culture media. The volatile oil mixture had a strong effect on colonies of enterobacteria, yeasts and moulds: 0 CFU. No colony developed compared to the comparison plates (standard). We mention that to test the antimicrobial activity, we did not encapsulate the mix of volatile oils in sodium alginate. We did this analysis last year, before doing the work specified above. We performed this analysis to choose the types of volatile oils to introduce into yoghurt. Please see: 236–242
Figure 1 Figure 2
Figure 1. Left: Petri dish on which enterobacteria culture was incorporated.
Right: Petri dish on which volatile oil + enterobacteria culture was incorporated.
Figure 2. Left: Petri dish on which mould culture was incorporated.
Right: Petri dish on which volatile oil + mould culture was incorporated.
- Simply consuming plants is not a controlled process. There may be certain toxic compounds in plants that develop from different natural causes, soil, climate, area. During the extraction of the compounds from the plants, a controlled process takes place. After obtaining the volatile oil, it is checked by gas chromatography and thus its main compounds can be determined. The oils obtained, once analyzed and the useful components identified, offer the possibility of their exact dosage in yoghurt so that the antioxidant effect can be ensured without substantially changing the taste and smell of the product. Please see: Row 158-164
In the introduction, we added some details about yoghurt and the health benefits to the consumer. Numerous studies conducted in recent years have shown that regular consumption of dairy products can have a protective effect against the development of obesity and cardiovascular disease. The most consumed dairy products are yoghurt and cheese. Yoghurt is defined as a food in the form of a thick and slightly sour liquid that is obtained by adding bacteria in milk. Yoghurt consumption has increased worldwide due to its nutritional value, therapeutic effects and functional properties. Due to these characteristics, we decided to choose yoghurt as the dairy product that we will enrich with antioxidant compounds. Please see: Row 54–61.
- Following the sensory analysis, the tasters stated that the volatile oil added to the yoghurt only slightly influences its taste. The specific taste of yoghurt is predominant, only at the end feeling a slight taste of the plant from which the volatile oil is extracted. Please see: Row 232-235.
- For each figure we added the units of antioxidant activity and the necessary explanations. Please see: Figure 1(A), 2 (A), 3 (A), 4 (A), 5 (A), 6.
- We detailed in the manuscript the method of extraction of volatile oils. For the extraction of volatile oils, we used mint, basil, lavender —dried and crushed aerial part— and fennel seeds. The volatile oil was extracted by steam entrainment using the Neo–Clevenger apparatus modified by Moritz. The extraction time was five hours for each sample, and for efficient extraction, the plants were soaked the day before. At the end of the extraction, the volatile oil obtained is measured and 1 mL of benzene is added over it. It is then placed in a glass vial containing sodium sulfate anhydrous to remove any traces of water. Using a Pasteur pipette, we extracted the volatile oil from the glass vial and the benzene evaporated. To preserve the characteristics of the volatile oil until analysis, it is sealed in a dark ampoule and stored in the refrigerator. Please see: Row 251-259.
We detailed in the manuscript the antioxidant analysis. For the extraction of the compounds to determine the antioxidant capacity of the yoghurt samples with volatile oils, we used the extraction method adapted after Patel et al. (2016). We weighed 0.5 g of the sample to be analyzed and then extracted it with 10 mL of the mixture Methanol: Water: Hydrochloric acid 0.12 M = 70 : 29 : 1 (v / v / v), at room temperature, for 24 hours. The mixture was then kept on the ultrasonic water bath for 30 minutes at a temperature of 25 °C. After the time runs out, the supernatant was collected and centrifuged at 8000 rpm for 10 minutes. The residue was suspended in 10 mL of solvent to perform a second extraction for 15 minutes, on the bath of water at 25 °C. The resulting supernatant was centrifuged under the same conditions as the first. The total amount of supernatant was evaporated to the rotary evaporator and the residue was taken up with 10 mL of methanol. We filtered the mixture of supernatant and methanol and filled it to a volume of 10 mL with the same solvent. To determine the antioxidant activity of the yoghurt samples with the addition of volatile oils encapsulated in sodium alginate, we used a method adapted according to the method applied by Tylkowski et al. (2011) for the ethanolic extracts of Sideritis ssp. L. We prepared a 25 µg / mL DPPH solution by solubilizing a quantity of DPPH in absolute methanol. This mixture needed to be prepared in advance —at least 1–2 hours— for complete solubilization. We weighed 970 µL DPPH solution 25 µg / mL and added over 30 µL methanol extract from the samples to be analyzed, which we obtained using the extraction shown above. To interpret the results, we measured the absorbance at 515 nm for each sample using the CECIL 1021 UV-VIS spectrophotometer and plotted an absorbance graph based on concentration. Please see: Row 264-283.
- The yoghurt samples were packed in 150g plastic cups and stored in the refrigerator at a temperature between 0–4 °C. During the entire storage period, the glasses were covered with aluminium foil. Please see: Row 217-219.
Thank you very much for your constructive comments and insightful suggestions on the manuscript.

Round 2
Reviewer 1 Report
The manuscript has been improved significantly and is not suitable for publication.
Note: Line 461 is labelled "6.Patents". This must be an error....
Author Response
Response to Reviewer 1 Comments
Point 1: Line 461 is labelled "6.Patents". This must be an error....
Response 1: We checked the manuscript again and it is indeed an error. We deleted the label "6.Patents"`and marked this with Track Changes (Please see: Row 460).
Thank you very much for your constructive comments and insightful suggestions on the manuscript.

Reviewer 2 Report
The manuscript has been significantly modified and improved compared to the initial submission.
The manuscript is internally consistent and is now sufficiently concise for publication.
On page 11, there is a section 6 labelled 'Patents' which is probably an error.
Author Response
Response to Reviewer 2 Comments
Point 1: On page 11, there is a section 6 labelled 'Patents' which is probably an error.
Response 1: We checked the manuscript again and it is indeed an error. We deleted the label "6.Patents"`and marked this with Track Changes (Please see: Row 460).
.
Thank you very much for your constructive comments and insightful suggestions on the manuscript.
